# Evaluation of Machine Leaning Algorithms for Streets Traffic Prediction: A Smart Home Use Case

**DOI:** 10.3390/s23042174

**Published:** 2023-02-15

**Authors:** Xinyao Feng, Ehsan Ahvar, Gyu Myoung Lee

**Affiliations:** 1Learning, Data and Robotics Laboratory, ESIEA Graduate Engineering School, 75005 Paris, France; 2Nokia, 91300 Massy, France; 3School of Computer Science and Mathematics, Liverpool John Moores University, Liverpool L3 3AF, UK

**Keywords:** machine learning, deep learning, recommendation system, energy consumption, smart home, neural network, performance evaluation

## Abstract

This paper defines a smart home use case to automatically adjust home temperature and/or hot water. The main objective is to reduce the energy consumption of cooling, heating and hot water systems in smart homes. To this end, the residents set a temperature (i.e., X degree Celsius) for home and/or hot water. When the residents leave homes (e.g., for work), they turn off the cooling or heating devices. A few minutes before arriving at their residences, the cooling or heating devices start working automatically to adjust the home or water temperature according to the residents’ preference (i.e., X degree Celsius). This can help reduce the energy consumption of these devices. To estimate the arrival time of the residents (i.e., drivers), this paper uses a machine learning-based street traffic prediction system. Unlike many related works that use machine learning for tracking and predicting residents’ behaviors inside their homes, this paper focuses on predicting resident behavior outside their home (i.e., arrival time as a context) to reduce the energy consumption of smart homes. One main objective of this paper is to find the most appropriate machine learning and neural network-based (MLNN) algorithm that can be integrated into the street traffic prediction system. To evaluate the performance of several MLNN algorithms, we utilize an Uber’s dataset for the city of San Francisco and complete the missing values by applying an imputation algorithm. The prediction system can also be used as a route recommender to offer the quickest route for drivers.

## 1. Introduction

Smart homes can make our lives more comfortable and exciting. Security, user privacy and energy consumption are considered as the three main challenges in smart homes.

In this paper, we focus on the energy consumption issue. Heating, ventilation, and air conditioning account for around half of building energy consumption in the U.S. and between 10 and 20% of total energy consumption in developed countries [1].

On the other hand, the number of vehicles in large-size smart cities has sharply increased in recent years. In large-size cities, traveling duration between two special locations can be varied depending on the time of day and even the day of week (i.e., we can have different traffic situations). This has made the traffic one of the smart city challenges. A short-term prediction for the street traffic situation of routes is not only essential for drivers (e.g., to save time and automotive fuel), but can also be used as useful information for various use cases and applications.

Putting all pieces together, we propose a smart home use case to automatically adjust home temperature and hot water temperature. Here, the street traffic prediction system predicts the arrival time of a home resident (i.e., who is driving) and, just a few minutes before the arrival, the cooling or heating devices will be turned on to reduce the energy consumption of the devices.

The system predicts the street traffic situation using a machine learning technique (model). In general, machine learning and neural network-based (MLNN) algorithms play an important role on the performance of machine learning-based systems.

However, the performance of an MLNN algorithm may depend on the application and/or the utilized dataset. While an MLNN algorithm can show a great performance for one application, it may not necessarily provide this level of performance for another application or even another dataset.

For this reason, we evaluate the performance of several famous MLNN algorithms to find which one is more appropriate for our application (i.e., traffic prediction). We evaluate the performance of the following MLNN algorithms: K-nearest neighbors (KNN), decision tree (DT), random forest (RF), support vector machines (SVMs), multilayer perceptron (MLP), long short-term memory (LSTM), single layer perceptron (SLP) and categorical naive Bayes (CNB).

In addition to MLNN algorithms, the utilized datasets play an important role in the street traffic prediction process and results. Even if the systems are efficient, certain limitations can occur because of the uncertainty in traffic-related data [2].

While some datasets are available for the traffic situation of several highways/freeways or some special streets of different cities, finding comprehensive and complete datasets covering all small and large-size streets of a city is still a big issue. As a solution, in this paper, we utilize an Uber dataset (which covers a large number of small streets) for the city of San Francisco and then complete the missing values by applying an imputation algorithm.

We use Scikit-Learn [3] and Pytorch [4] to implement the algorithms and consider the following evaluation metrics: Precision, F1-Score, and Accuracy.

We summarize our contributions as follows:Propose a use case for smart home to adjust the home temperature and hot water temperature by controlling the heating and cooling devices with the main goal of reducing the energy consumption of the devices;Propose a dual-objective machine learning-based streets traffic prediction system. It estimates a resident (i.e., a driver) arrival time and sends it (i.e., as a context) to the context-aware smart home application. It can be also used as a route recommender to guide the quickest routes for drivers;Implementing several famous MLNN algorithms for the streets’ traffic prediction;Analyzing and comparing the performance of the implemented MLNN algorithms and introducing the most appropriate one for the problem of streets traffic prediction.

The rest of the paper is organized as follows. The related work is presented in Section 2. Section 3 introduces our use case. An overview of the implemented and analyzed algorithms in this paper is presented in Section 4. Section 5 evaluates the performance of several famous MLNN algorithms to find the most appropriate one for our application. We will have a discussion and conclusion in Section 6 and Section 7, respectively.

## 2. Related Work

### 2.1. Machine Learning for Smart Homes

Machine learning has been used for various applications in smart homes. For example, Taiwo et al. [5] presented a system to control home appliances, monitor environmental factors, and detect movement in the home and its surroundings. They utilized a deep learning model for motion recognition and classification based on the detected movement patterns. Using a deep learning model, they designed a system for intruder detection. Based on the walking pattern, a human detected by the surveillance camera is categorized as an intruder or home residence.

Filipe et al. [6] presented a voice-activated smart home controller. They proposed an architecture that focuses on the use of Online Learning to develop a smart home controller capable of controlling multiple connected devices according to the resident’s preferences and habits.

Fahim et al. [7] proposed ApplianceNet to detect daily life activities in smart homes. It is based on the energy consumption patterns of home appliances attached to smart plugs. They use a multi-layer, feed-forward neural network to classify the home appliances.

Machine learning can also help smart homes reduce energy consumption.

Fakhar et al. [8] recently provided a survey of smart home energy conservation techniques. They identified various critical features in energy conservation techniques (e.g., user and appliance energy profiling) to perform a comparative analysis among various techniques. They also presented various energy conservation techniques and provided a statistical analysis of the existing literature.

Kim et al. [9] proposed a temperature controller based on machine learning. The system learns the life patterns and desired temperature of individual residents (according to each situation) and allows the learning results to be reflected in the temperature control (i.e., cooling/heat devices).

All the aforementioned machine learning-based works have mainly focused on tracking and predicting the home resident’s mobility patterns or behavior inside homes. In contrast, our work uses machine learning to predict the arrival time of a resident based on the predicted (route) traffic situation.

### 2.2. Streets Traffic Prediction

Several approaches have been proposed to predict the traffic situations of streets and highways. Street traffic prediction approaches can be divided into two main categories: parametric (e.g., exponential smoothing, the Kalman filtering [10] and ARIMA [11]) and non-parametric (e.g., machine learning-based approaches).

The parametric approaches estimate the traffic situation based on the strong theoretical assumption. The studies show that non-parametric approaches outperform parametric solutions because of their ability to deal with a large number of parameters and big data [2,12].

Jose Braz et al. [13] developed and compared three deep learning models for forecasting the traffic flow in the Barra and Costa Nova regions. They divided their dataset into training, validation, and testing sets (i.e., the holding out method).

Kim et al. [14] presented a structural RNN architecture that combines the road network map with the traffic speed data to predict the future traffic speed. They used a traffic speed dataset from the case studies of the SETA EU project [15].

In contrast to the mentioned works, we utilize the walk forward validation. For the time series data analysis, the walk forward validation can offer a better result than holdout and k-fold cross-validation [16].

In addition, one common challenge for the existing related work is that the traffic road datasets under their study only cover several inter-city highways or the main avenues of a city or a smaller number of streets which are most of the time sparsely located. When we want to predict the traffic situation of every requested route (i.e., in source–destination pair format) in a city, these datasets are not very useful. To solve this problem, we used an Uber dataset for the city of San Francisco and completed the missing values by applying an imputation algorithm (see details in Section 5).

## 3. Use Case

In this paper, we present a use case to automatically adjust one’s home temperature (and/or hot water temperature) according to the desire temperature which has been already set by the resident. The main objective is to reduce the energy consumption of the cooling or heating devices in smart homes. When the residents are not at home, the cooling or heating devices should be off. A few minutes before a resident returns, the devices start working to adjust the requested temperature. To this end, it is necessary to know the estimated arrival time of the residents. This use case is more useful for the resident who spends a considerable duration of time outside their home. Notice that calculating the needed time to adapt the home temperature degree (or water temperature) according to a resident request is possible. This mainly depends on several factors such as the home size, number and power of the cooling and heating devices at home, outside temperature and requested temperature degree. However, this is out of scope of our work (i.e., this paper).

### 3.1. Smart City

We consider a smart city including a number of intersections. Here, an intersection is a place or point where two or more streets cross each other. A street (or a part of that) which is located between two intersections is considered as a segment (see Figure 1).

### 3.2. Scenario

We present the use case considering the following scenario. Assume that a driver is now somewhere in a large-size city and they decide to return home. We estimate the arrival time of the driver and send it to the related home application (i.e., here, the application of adjusting home temperature) to adjust the home temperature based on the arriving time of the driver.

We assume that every driver has a list of source/destination pairs and selects an appropriate source/destination pair from the list. Here, the source is the closest intersection to the current location of the driver and the destination is the closest intersection to their home.

We propose a system that receives a driver request (i.e., including the leaving time and their selected source/destination pair) and recommends the best routes to the driver and the estimated arrival time to the smart home application.

As Figure 2 shows, we define 12 steps to better present the use case and working mechanism of the system. In step 1, a driver (i.e., a home resident which is driving somewhere in the city now) sends a route request to the system including a source/destination pair and leaving time.

When the system (i.e., central controller) receives the request, it selects the four shortest routes between the requested source and destination with the help of the route finder component (i.e., steps 2 and 3). In fact, the route finder component creates a graph where the intersections are its vertices and the segments are its edges. Having this graph, it can find the four shortest routes between every requested source and the destination using a modified version of Dijkstra’s algorithm (or any other shortest route/path finder algorithm). Every selected route consists of a set of segments. For every segment of a selected route, we have historical data (introduced in Section 5.1). The system applies an MLNN algorithm to the historical data of every segment of selected routes one-by-one to predict the traffic situation of them for the requested time interval (i.e., steps 4 and 5).

To this end, our system considers three levels of (average) car movement speeds for segments called L1 (heavy traffic), L2 (normal traffic), and L3 (low traffic) considering two thresholds (i.e., F1 and F2) as follows:The average speed less than or equal to F1 is defined as heavy traffic or level L1;The average speed between F1 and F2 is defined as normal traffic or level L2;The average speed equal or above F2 is defined as low traffic or level L3.

Here, the classification technique for traffic prediction gives us the opportunity to send simple and easy-to-understand information to drivers (i.e., sending traffic information of every segment as low, normal or heavy).

In the next steps (i.e., steps 6 and 7), the system calculates the length of every segment of the selected routes utilizing the segment length calculator component. Recall that the system has the geographical information of intersections and segments.

In steps 8 and 9, having the length and traffic class (i.e., low, normal, or heavy) information of the segments of every selected route, the system estimates the traveling duration of the four selected routes.

In step 10, the system suggests the selected routes, the traveling duration of every selected route and the information related to segments of every selected route (i.e., traffic level) to the requested driver. In step 11, the driver selects one of the four suggested routes and informs the system about it. Finally, our system sends the estimated arrival time of the driver to the smart home application (i.e., as an external context) in step 12. Here, we assume that the system can communicate to both drivers and smart home applications. The smart home application knows the current temperature of the home (i.e., using the home temperature sensors). After receiving the estimated arrival time, if necessary, the application sends appropriate control signals to turn the cooling or heating devices on.

Recall that this paper mainly focuses on the benefits of combining smart home use cases and the street traffic prediction system (i.e., here, the arrival time predictor system) rather than going into the details of the smart home applications. As already mentioned, this traffic prediction system can also help a home residence (i.e., when driving) to select the most appropriate route. The traffic prediction system (i.e., the arrival time predictor system) can be installed on the Cloud and provide the service to several drivers.

## 4. Machine Learning and Neural Network Algorithms

As we already mentioned, we cannot find an MLNN algorithm that works perfectly for all types of scenarios, applications and datasets. In this case, we need to study and evaluate several well-known MLNN algorithms to find which one can be more appropriate for our work. This section briefly presents MLNN algorithms that we analyze in this paper.

### 4.1. K-Nearest Neighbors (KNN)

KNN is a simple and lazy learning algorithm that can be used for both classification and regression. It stores all the available data and classifies a new data point based on a similarity measure (e.g., Euclidean distance function). In other words, it does a simple majority vote of its k-nearest neighbors for classifying a new data. Detecting an optimal number of neighbors for a new data plays an important role in the performance of KNN [17].

### 4.2. Decision Tree (DT) and Random Forest (RF)

Similar to KNN, DT can be utilized for both classification and regression tasks. It uses a hierarchical tree-like structure where it sets different conditions on the branches. Iterative Dichotomiser 3 (ID3), C4.5, and Classification and Regression Trees (CART) are well known DT algorithms. RF is another well-known machine learning algorithm. It is an ensemble classification technique consisting of many DTs. In other words, the forest is an ensemble of independent DTs in a way that every tree sets conditional features differently. After receiving a sample, the root node sends it to all the sub-trees. In the next step, each sub-tree predicts the class label for the sample. Finally, the class in the majority is given to that sample [17,18,19].

### 4.3. Support Vector Machines (SVMs)

SVM is considered a supervised learning algorithm to solve classification and regression prediction problems based on statistical learning theory [20]. In fact, it maps data points to high-dimensional vectors. For data points in an n-dimensional space, a (n−1)-dimensional hyperplane is considered as a classifier [21].

### 4.4. Categorical Naive Bayes (CNB)

Naive Bayes is a simple algorithm for classification dependent on Bayes’ theorem with a supposition of freedom among predictors. In CNB, it is naively considered that the features are independent [21,22,23].

### 4.5. Single-Layer Perceptron (SLP)

SLP is a binary classifier consisting of a node (called artificial neuron or perceptron) with a set of inputs and an output. SLP considers a weight for every input of the perceptron. It then multiplies the inputs with their respective weights. After that, the results of the products are added together. The calculated value is then compared with a threshold value to detect the (output) label [21].

### 4.6. Multilayer Perceptron (MLP)

MLP is a neural network algorithm. It can use several perceptrons where these perceptrons are distributed in at least three layers (i.e., one input layer, one or more hidden layers, and one output layer). MLP is usually more powerful and complex than SLP [21,24].

### 4.7. Long Short-Term Memory (LSTM)

LSTM is considered an improved version of the recurrent neural network (RNN) [25]. It is proposed to expand the RNN memory. This memory can keep information over an arbitrary length of time. Three gates (i.e., input, output, and forget gates) control the information flow into and out of the neuron’s memory [26].

## 5. Performance Evaluation

In this section, we first present the utilized dataset and the way of preparing it. After that, we present technical details for the implementation including the utilized tools, hyper-parameters, utilized features, validation technique and evaluation metrics. Finally, the third part describes how we implemented MLNN algorithms by the introducing main utilized Python functions.

### 5.1. Dataset Description

We utilized an Uber traffic dataset for San Francisco [27]. We selected a zone covering a total number of 439 segments in San Francisco. Traffic observations were recorded in approximately 1-hour intervals for each street segment during 18 h per day (i.e., from 6 AM to 24) for 5 weeks. The real dataset contains around 197,496 records. It then becomes equal to 276,570 records after calculating and filling its missing values (i.e., by applying the imputation algorithm. Details in Section 5.1.1).

Every row in the dataset refers to the situation of a segment at a specific timestamp. The columns correspond to the following content:date: the date (year–month–day) of the record;hour: the time (in hours) of the record (local city time);segment_id: shows ID of the corresponding street segment;osm_start_node_id: starting node (intersection) of the segment, ID of the node according to OpenStreetMap;osm_end_node_id: ending node of the segment, ID of the node according to OpenStreetMap;mean_speed: average speed of Uber vehicles during the time interval in this street segment in km/h;traffic_class: shows traffic situation of the segment (i.e., low, normal, heavy) during the time interval.

#### 5.1.1. Data Cleaning

We found that the obtained dataset contains several missing lines (data) where the observations are not regularly and continuously recorded every 1 h for each segment during the considered 5 weeks.

In order to solve this problem, we defined a two-step solution: step (1) dataset size reduction (to remove unnecessary lines/records); step (2) dataset completion (to add necessary/utilized missing data).

step 1—dataset size reduction: we found that a majority of the missing values were from the night. It is normal. Because Uber does not have a lot of services during the night (i.e., between 00:00 and 6:00). On the other hand, we know that we usually do not have heavy traffic between 00:00 and 6:00 and, therefore, it is not necessary to use the traffic recommendation system during these hours. As a result, we reduced the dataset size. Instead of considering observations for 24 h per day, we consider a dataset file keeping observations for 18 h per day (i.e., recorded observations between 6:00 and 24:00). This could remove a large number of missing values from our dataset (approximately 7.44 percent);step 2—dataset completion: in order to find appropriate values for the missing parts and complete the dataset, we use the Bayesian temporal matrix factorization (BTMF) model recently proposed in [28].

As an example, Figure 3 shows a part of the initial dataset for segment 0 (on 5 January 2020, from 6:00 to 24:00) with two missing values. Figure 4 shows that two missing values were calculated in step 2 (i.e., using BTFM).

#### 5.1.2. Dataset Individualization

We separated our dataset into 439 sub-datasets where each new dataset contains traffic observations for only one individual segment. The dataset separation helped us reduce the complexity and processing time and improve the performance. Recall that, according to the use case under study (see Section 3), we can predict the traffic situation for every segment of a selected route separately.

### 5.2. Implementation Technical Details

#### 5.2.1. Tools and Software

We utilized the Scikit-learn library [3] to implement the classic machine learning algorithms and PyTorch [4] for the deep learning algorithm.

#### 5.2.2. Input Features

Considering the dataset information and requirements of our use case, the following input features are defined and extracted:segment_id: shows ID of the corresponding segment;observation time: keeps the time (hour) of each observation;traffic level/class: is calculated based on the average speed vehicles in every segment. For defining the three traffic levels/classes, F1 and F2 threshold values are considered 10 and 30, respectively. The values are selected after doing several tests with different values. In this case, the average speed:−Less than or equal to 10 km/h is considered as level L1 (heavy traffic);−Between 10 and 30 is considered as level L2 or normal traffic;−Equal or above 30 is considered as level L3 or low traffic.

#### 5.2.3. Hyper-Parameters

Hyper-parameters play an important role in performance of the created models.

We use the Scikit-learn’s RandomSearch cross-validation [29] to find appropriate values for hyper-parameters.

Recall that CNB does not have hyper-parameters to tune.

One week (i.e., 20%) of the dataset is used for tuning the hyper-parameters. The obtained hyper-parameter values are shown in Table 1 and Table 2.

#### 5.2.4. Validation Technique

We know that our data (dataset) constitutes a time series. As some usual techniques such as the k-fold cross-validation technique do not work perfectly for time series datasets [16], we utilize the walk-forward validation technique (i.e., a technique preserving the order of data).

We separate our dataset into five equal units or partitions (i.e., a unit is equal to one week). The units are chronologically ordered. We considered the last four consecutive weeks of the dataset (i.e., 80% of the dataset) for training and testing our models. In each execution, we consider all the available data before the last unit as the training set and the last unit itself as our testing set.

As shown in Figure 5, for the first execution (i.e., Execution 1), we consider the W1 unit for training and W2 for testing. We then consider W1 and W2 units together for training and W3 for testing in the second execution. Finally, for the last execution (i.e., Execution 3), W1, W2, and W3 units are considered for training and W4 for testing.

Finally, the model scores (e.g., accuracy) are calculated as the average results of all executions [16,30].

#### 5.2.5. Evaluation Metrics

In this section, we evaluate the performance of the following MLNN algorithms: KNN, DT, RF, SVMs, MLP, LSTM, SLP and CNB.

In order to evaluate the aforementioned MLNN algorithms, we consider the accuracy, precision and F1-score metrics as well as the execution time of algorithms (i.e., including both training and testing times).

### 5.3. Implementation and Evaluation

We used Scikit-learn built-in machine learning classifiers (e.g., DecisionTreeClassifier for DT algorithm) to implement algorithms (except LSTM). The random search function available in Scikit-learn was utilized to detect appropriate values for hyper-parameters.

For every algorithm, we calculated its training/testing execution time by measuring the starting time and ending time of the execution. To do this, we utilized the available “time” function. As we used the walk forward methodology where we executed an algorithm several times (with different training and testing sizes), we calculated the execution time of an algorithm as an average of all execution times of that algorithm.

Notice that we utilized the one-vs.-one method for using binary classification algorithms such as SLP for our three-class classification.

Table 3 shows the average scores of the algorithms (models) and their training and testing times (in seconds) given 1 h predictions.

By analyzing Table 3, we extracted the best algorithms in Table 4. We can observe that RF could outperform other algorithms in accuracy and F1_score. It is also the second best algorithm in terms of precision. However, for the training and testing times, it is one of the worst algorithms. This is because it is an ensemble learning algorithm.

CNB can be considered the second best algorithms (i.e., ranked 2 in accuracy and testing time and ranked 3 in precision, training time and F1_score).

DT shows promising results. It is ranked among the top three algorithms in accuracy, F1_score, training time and testing time. Its precision is equal to 90.10 which makes it the fourth algorithm in the table.

We can see that SVM and KNN are ranked in the middle of the table. While the results of SVM are slightly better than KNN in terms of accuracy and precision, KNN has a slightly better results for testing and training times.

The neural network-based algorithms (i.e., SLP, MLP and LSTM) have the worst results in accuracy and F1_score. While the training time for MLP and LSTM, as two complex algorithms, is rather long, SLP has a rather short training time.

Putting all the pieces together, RF has the best cross-validation scores. However, as the results show, a complex algorithm such as RF is time- and energy-consuming. As reducing energy consumption is the main objective of our work, our priority is to select and use a simpler and quicker algorithm (having a performance close to that of RF) such as CNB or even DT for using in our use case.

## 6. Discussion

Energy efficiency (cost) is an important factor for smart home applications. Many studies have been performed to improve the energy consumption of smart home applications using machine learning. However, they generally focus on tracking and predicting resident behavior in smart homes and use it as a context for applications. In contrast, our work shows that machine learning can also help smart home applications improve energy consumption by considering and predicting a resident’s behavior outside a smart home (i.e., as an external context for smart home systems and applications). The proposed street traffic prediction system can be also used as a route recommender (i.e., to save time and automotive fuel) when the residents are not home or as an assistant in other use cases and applications.

To find the most suitable MLNN algorithm for traffic prediction, we evaluated the performance of a set of well-known MLNN algorithms (i.e., KNN, DT, RF, SVM, MLP, LSTM and SLP) based on (a modified version of) a real Uber parking dataset in San Francisco. Although the RF algorithm shows the best performance, its training is very time- and energy-consuming. On the other hand, we can see that the performance of some simple algorithms such as CNB and DT is very close to RF. In this situation, it can be a good idea to use a simple algorithm with a lower execution time.

For traffic prediction using machine learning, there are still many open research problems and challenges that have not been fully investigated. Finding appropriate solutions to consider abnormal conditions (e.g., extreme weather or temporary traffic control) is one of the current challenges. Benchmarking traffic prediction, the interpretability of models, real-time prediction and choosing an optimal network architecture are additional issues in using machine learning for traffic prediction [31].

## 7. Conclusions

We proposed a smart home use case (i.e., temperature adjustment by controlling the heating and cooling systems automatically). The main goal was to reduce energy consumption of the cooling and heating systems. A machine learning-based traffic prediction system was designed for the use case to estimate residents’ arrival time based on the predicted traffic situation. It can also recommend residents what appropriate routes to take when they are in the city and want to return home. Unlike many works that use machine learning to track and predict the residents behavior inside their homes, we used machine learning to predict the residents behavior (i.e., arrival time) when they are outside their smart home (i.e., driving). We believe that predicting residents behavior outside smart homes can also help to improve energy consumption of some smart home applications. We also found that some simple algorithms (e.g., CNB and DT) can provide a very close performance to RF performance with a better execution time.

## Figures and Tables

**Figure 1 sensors-23-02174-f001:**
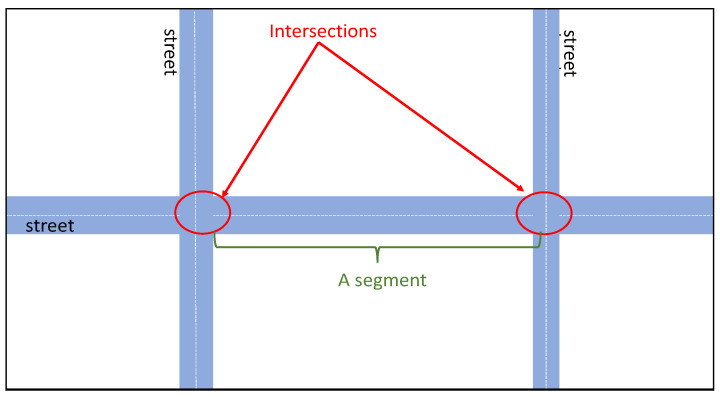
Intersection and segment concepts.

**Figure 2 sensors-23-02174-f002:**
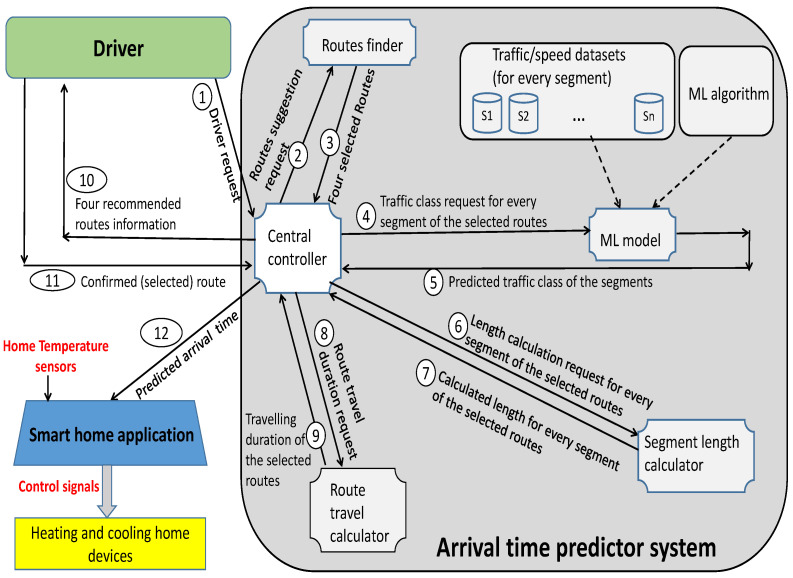
Working mechanism of the proposed system.

**Figure 3 sensors-23-02174-f003:**
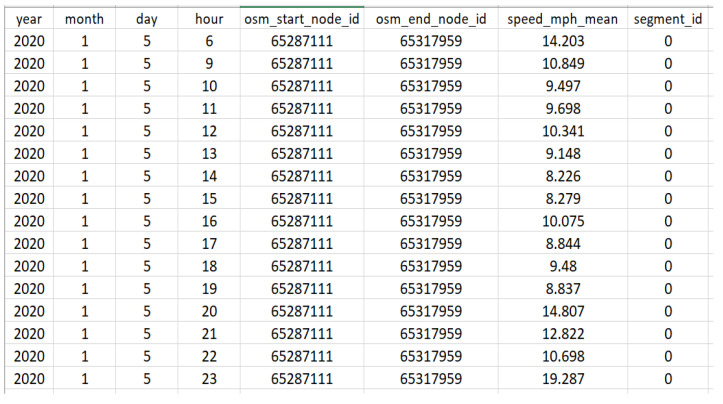
An illustration of the initial dataset for segment 0 (on 5 January 2020, from 6:00 to 24:00) with two missing values.

**Figure 4 sensors-23-02174-f004:**
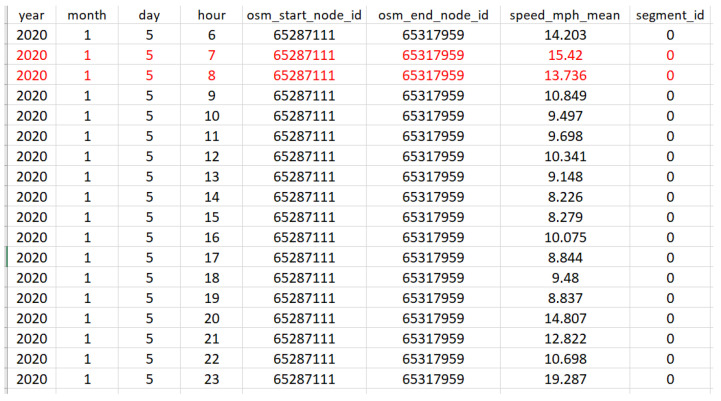
An illustration of the final dataset for segment 0 (5 January 2020, from 6:00 to 24:00) where the two missing values (shown in red color) were calculated using BTFM.

**Figure 5 sensors-23-02174-f005:**
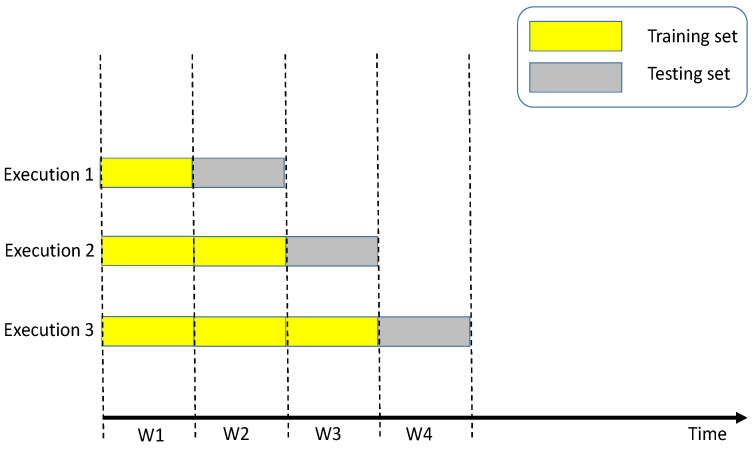
The walk-forward working mechanism for 5 weeks (units).

**Table 1 sensors-23-02174-t001:** Obtained values for hyper-parameters—Part I.

KNN		DT		RF	
Parameter	Value	Parameter	Value	Parameter	Value
n_neighbors	14	max_depth	8	max_depth	19
metric	manhattan	criterion	entropy	criterion	gini
weights	distance	min_samples_leaf	3	min_samples_leaf	4
				n_estimators	270

**Table 2 sensors-23-02174-t002:** Obtained values for hyper-parameters—Part II.

SVM		SLP		MLP		LSTM	
**Par.**	**Val.**	**Par.**	**Val.**	**Par.**	**Val.**	**Par.**	**Val.**
kernel	rbf	loss	perceptron	hidden_size	130	input_size	2
C	100	alpha	100.0	solver	adam	hidden_size	20
gamma	1	penalty	l2	activation	logistic	layer_dim	1
-	-	learning_rate	adaptive	learning_rate	invescaling	-	-
-	-	eta0	0.001	-	-	-	-

**Table 3 sensors-23-02174-t003:** Average cross-validation scores of models and their training and testing times.

Metrics	CNB	DT	KNN	MLP	RF	SLP	SVM	LSTM
Accuracy	90.09	89.81	89.20	87.34	90.53	83.89	89.75	81.55
Precision	90.58	90.10	89.38	89.09	90.99	83.43	89.64	93.17
F1_score	88.31	88.72	88.25	84.03	89.10	82.93	88.72	77.10
Training	0.0013	0.0012	0.0011	0.2334	1.0843	0.0013	0.0021	0.7882
Testing	0.0008	0.0010	0.0016	0.0016	0.0172	0.0006	0.0017	0.0006

**Table 4 sensors-23-02174-t004:** The best algorithms according to different metrics.

Accuracy	F1_Score	Precision	Training	Testing
RF(90.53)	RF(89.10)	LSTM(93.17)	KNN(0.0011)	LSTM, SLP(0.0006)
CNB(90.09)	DT/SVM(88.72)	RF(90.99)	DT(0.0012)	CNB(0.0008)
DT(89.81)	CNB(88.31)	CNB(90.58)	CNB, SLP(0.0013)	DT(0.0010)

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
