# Peer review of "Evaluation of Machine Leaning Algorithms for Streets Traffic Prediction: A Smart Home Use Case"

_sensors, 2023, doi:10.3390/s23042174_

Round 1
Reviewer 1 Report
Sensors
The authors present the application of machine learning (ML) in street traffic prediction, targeting smart home use cases. Emphasizing QoS and the experimental results indicated that the commonly used ML algorithms work well in the study. The research is interesting (combining smart home use cases and traffic prediction). However, the quality of the current form of the manuscript is very weak and needs major improvement.
- The manuscript is poor presented and hard to follow
- No significant contribution and authors should elaborate more on some other aspects. This can be done by writing a deep analysis of the mechanism in every ML algorithm on how they affected the results.
- Is QoS the main contribution? Should highlight more on QoS in the title and related works
- Problem background. What are current research problems on the smart home use case and traffic prediction using ML?
- Many grammatical errors. Need proofreading from professional editors.
- Give a snapshot of the dataset before and after pre-processing (cleaning).
- Line 248: What do you mean one week of the dataset is used for tuning the hyper-parameters? Why are Table 1 and table two separated by Part?
- Give brief and short descriptions of KNN, DT, RF, SVMs, MLP, LSTM, SLP and CNB so the reader can better understand how each of the ML algorithms works.
- Table 3: Highlights / bold the most important values.
- Is there any significant finding on execution time?
- The result shows that simple algorithms work well and are close to RF. Meaning there is weak novelty and applicability.
- Authors should discuss more on the data itself.
- Again, the research background is interesting, but the current form of the manuscript is far from reaching the standard of Sensor journal.

Author Response
# Reviewer 1
The authors present the application of machine learning (ML) in street traffic prediction, targeting smart home use cases. Emphasizing QoS and the experimental results indicated that the commonly used ML algorithms work well in the study. The research is interesting (combining smart home use cases and traffic prediction). However, the quality of the current form of the manuscript is very weak and needs major improvement.
− The manuscript is poor presented and hard to follow
Answer: To improve the presentation of the paper:
- We re-wrote the abstract section to clarify the main objective of the work
- We revised the key parts of the document (i.e., contribution description in Introduction section, scenario description in Use Case section, Figure 2 modified)
- A section has been added to the document to briefly describe the ML algorithms that we implemented in this work.
− No significant contribution and authors should elaborate more on some other aspects. This can be done by writing a deep analysis of the mechanism in every ML algorithm on how they affected the results.
Answer: The main novelty of this work is the idea of using traffic prediction information (as a context) for context-aware smart home applications. We added more description to the performance evaluation section and considered a ranking system for better analyzing the algorithms.
− Is QoS the main contribution? Should highlight more on QoS in the title and related works
Answer: The main contribution (let say objective) is energy saving (i.e., reducing energy consumption of the cooling/heating devices at home) while still satisfying the residents requirement (i.e., their requested temperature when they arrive).
To reach the above mentioned objective, our contributions are:
- Proposing a use case for smart home to adjust home temperature by controlling the heating and cooling devices with the main goal of reducing energy consumption of the devices.
- Proposing a dual-objective machine learning-based streets traffic prediction system. It estimates a resident (driver) arrival time and sends it (i.e., as a context) to the smart home application. It can be also used as a route recommender to guide the quickest routes to drivers.
- Implementing several famous MLNN algorithms for the streets traffic prediction.
- Analyzing and comparing the performance of the implemented MLNN algorithms and introducing the most appropriate one for the problem of streets traffic prediction.
− Problem background. What are current research problems on the smart home use case and traffic prediction using ML?
Answer: Energy saving is considered as one of the most important research problems for smart homes. Designing control systems to manage home devices and reduce its energy consumption is a hot topic. ML has been used in smart home control systems to track and predict residents location or predict their behaviors. Unlike related works, we use ML to analyze the residents behavior outside homes and use it as a context for the smart home application to reduce energy consumption.
− Many grammatical errors. Need proofreading from professional editors.
Answer: improved.
− Give a snapshot of the dataset before and after pre-processing (cleaning).
Answer: done.
− Line 248: What do you mean one week of the dataset is used for tuning the hyper-parameters? Why are Table 1 and table two separated by Part?
Answer: Dataset includes five weeks. We separated the first week of the dataset (i.e., the data that gathered in the first week) and used it to find optimal values for hyper-parameters. It means the dataset that we used for finding hyper-parameter values is different from the dataset that we used for training and testing our model.
Tables 1 and 2 separated by part just to improve readability (considering the template format of the journal, we preferred to have two simple tables).
− Give brief and short descriptions of KNN, DT, RF, SVMs, MLP, LSTM, SLP and CNB so the reader can better understand how each of the ML algorithms works.
Answer: It is expected the readers have knowledge about these well-known algorithms. Looking at several works, they also do not describe the algorithms. However, we added a section to the document to briefly describe every algorithm.
− Table 3: Highlights / bold the most important values.
Answer: We added a part to the document where we introduced the three first/best algorithms for every metric.
− Is there any significant finding on execution time?
Answer: It proves that complex algorithms such as RF (ensemble learning) are time consuming. It is important in aspect of energy consumption.
− The result shows that simple algorithms work well and are close to RF. Meaning there is weak novelty and applicability.
Answer: 1) We did not mention that results of “all” simple algorithms are close to RF. For example, there is a big difference between SLP (as a simple single-neuron algorithm) and RF results.
2) The results are only valid for this application application and according to our dataset (therefore, we can not generalize our results for every application and dataset).
3) Meaning of the “novelty” that you used here is not very clear for us. In this part, our goal was to implement and compare results of different algorithms to detect which one is the most appropriate one for our work. Therefore, we did not look for any “novelty”.
− Authors should discuss more on the data itself.
Answer: As mentioned, we added two figures to show how the dataset is prepared (i.e., the missing values have been estimated). Recall that we already mentioned the features of the original dataset and the extracted features, dataset source etc.
− Again, the research background is interesting, but the current form of the manuscript is far from reaching the standard of Sensor journal

Reviewer 2 Report
The article is well written, methodology is clear and reproducible, and results are clear.

Author Response
# Reviewer 2
The article is well written, methodology is clear and reproducible, and results are clear.
Answer: Thanks.

Reviewer 3 Report
This paper designs a street traffic prediction system based on machine learning and applies it to the use case of smart home. The system is able to estimate residents' arrival times and plan an optimal route for drivers based on predicted traffic conditions. By evaluating the performance of several algorithms based on machine learning and neural networks (MLNN), the most suitable one is selected for the proposed system.
It is innovative to help improve QoS and energy consumption of smart home applications by predicting resident behavior outside of smart home.
Problems and suggestions:
1. It is mentioned in the Introduction that the performance of an MLNN algorithm may depend on the application and/or utilized dataset,but the paper only used the dataset Uber for training. I suggest the author add more data sets for comparative experiments.
2. In the step 2 of data cleaning, what algorithm is used for data interpolation?
3. Why do the authors attribute the training task to a classification task instead of a regression task? In other words, why do the authors predict traffic flow categories instead of directly predicting traffic flow? Please explain the problem.
4. The authors use the Walk Forward approach, in which an algorithm is executed multiple times using training and test sets of different sizes. Please add more details about the size of the training and test sets.
5. The authors use four indicators (accuracy, precision, recall, and f1-score) to evaluate the performance of each algorithm, but only analyze the results of the accuracy indicator. It is suggested that the authors supplement the analysis of the results of the other three indicators.
6. The authors should check the manuscript carefully since there are some grammatical errors in English. For example, in lines 223-226, “Since, according to the use case under study (see Section 3), we should predict the traffic situation of every segment of a selected route individually, this dataset separation can help us to reduce complexity, processing time and improve performance”. There is an obvious grammatical error in this sentence, which is the existence of two predicate verbs. Additionally, in the manuscript, punctuations are missing in several places.
7. The authors should analyze more related new works, e.g. "Exploring Human Mobility for Multi-Pattern Passenger Prediction: A Graph Learning Framework." IEEE Transactions on Intelligent Transportation Systems 23, no.(9), pp.16148–16160, 2022.

Author Response
# Reviewer 3
- It is mentioned in the Introduction that the performance of an MLNN algorithm may depend on the application and/or utilized dataset,but the paper only used the dataset Uber for training. I suggest the author add more data sets for comparative experiments.
Answer: It is true that there are some other datasets for the street traffic prediction. But they only cover a few main streets or highways which can not be appropriate for our work (i.e., routing). That’s why we preferred to prepare and use comprehensive datasets (i.e., covering many streets) for our work rather than using several datasets which cover only a few streets. Although we agree with you that using several datasets can be better, evaluating the algorithms with one dataset is common as a majority of the related work considered only one dataset to evaluate performance of the algorithms.
- In the step 2 of data cleaning, what algorithm is used for data interpolation?
Answer: As we already mentioned in the document, we used BTFM. For details, you can see reference 5.
- Why do the authors attribute the training task to a classification task instead of a regression task? In other words, why do the authors predict traffic flow categories instead of directly predicting traffic flow? Please explain the problem.
Answer: The classification technique for traffic prediction gives us the opportunity to send simple and easy-to-understand information to drivers (i.e., sending traffic information of every segment as low, normal or heavy).
- The authors use the Walk Forward approach, in which an algorithm is executed multiple times using training and test sets of different sizes. Please add more details about the size of the training and test sets.
Answer: We separate our dataset into 5 equal units or partitions (i.e., a unit is equal to one week). As shown in Figure. 3, for the first execution (i.e., Execution 1), we consider W1 unit for training and W2 for testing. We then consider W1 and W2 units together for training and W3 for testing in the second execution. Finally, for the last execution (i.e., Execution 3), W1, W2 and W3 units are considered for training and W4 for testing.
We added the above description to the document.
- The authors use four indicators (accuracy, precision, recall, and f1-score) to evaluate the performance of each algorithm, but only analyze the results of the accuracy indicator. It is suggested that the authors supplement the analysis of the results of the other three indicators.
Answer: To better present our analysis, we combined two result tables and added a clear and comprehensive description to the document.
- The authors should check the manuscript carefully since there are some grammatical errors in English. For example, in lines 223-226, “Since, according to the use case under study (see Section 3), we should predict the traffic situation of every segment of a selected route individually, this dataset separation can help us to reduce complexity, processing time and improve performance”. There is an obvious grammatical error in this sentence, which is the existence of two predicate verbs. Additionally, in the manuscript, punctuations are missing in several places.
Answer: done, thanks.
- The authors should analyze more related new works, e.g. "Exploring Human Mobility for Multi-Pattern Passenger Prediction: A Graph Learning Framework." IEEE Transactions on Intelligent Transportation Systems 23, no.(9), pp.16148–16160, 2022.
Answer: several recently published references have been added.

Round 2
Reviewer 1 Report
The paper have been revised based on the previous comments. Some are not well address:
1) What are current research problems on the smart home use case and traffic prediction using ML?
2) Line 379-384: Should present in table
Author Response
The paper have been revised based on the previous comments. Some are not well address:
1) What are current research problems on the smart home use case and traffic prediction using ML?
We have added the related descriptions for research challenges in the Introduction and discussion sections.
2) Line 379-384: Should present in table
We have made the table.
Reviewer 3 Report
The paper has been revised accordingly.
Author Response
The paper has been revised accordingly.
We very appreciate your review.